# Synthesis and Biological Activity of Novel α-Conotoxins Derived from Endemic Polynesian Cone Snails

**DOI:** 10.3390/md21060356

**Published:** 2023-06-09

**Authors:** Yazid Mohamed Souf, Gonxhe Lokaj, Veeresh Kuruva, Yakop Saed, Delphine Raviglione, Ashraf Brik, Annette Nicke, Nicolas Inguimbert, Sébastien Dutertre

**Affiliations:** 1CRIOBE, UAR CNRS-EPHE-UPVD 3278, Université de Perpignan Via Domitia, 58 Avenue Paul Alduy, 66860 Perpignan, France; yazid.souf@univ-perp.fr (Y.M.S.); delphine.raviglione@univ-perp.fr (D.R.); 2Faculty of Medicine, Walther Straub Institute of Pharmacology and Toxicology, Ludwig Maximilian University of Munich, Nußbaumstraße 26, 80336 Munich, Germany; g.lokaj@lmu.de (G.L.); annette.nicke@lrz.uni-muenchen.de (A.N.); 3Schulich Faculty of Chemistry, Technion-Israel Institute of Technology, Haifa 3200008, Israel; veeresh1590@gmail.com (V.K.); vkyakop.saed@campus.technion.ac.il (Y.S.); abrik@technion.ac.il (A.B.); 4IBMM, Université Montpellier, CNRS, ENSCM, 34093 Montpellier, France

**Keywords:** conotoxin, peptide synthesis, two-electrode voltage clamp, nicotinic acetylcholine receptors

## Abstract

α-Conotoxins are well-known probes for the characterization of the various subtypes of nicotinic acetylcholine receptors (nAChRs). Identifying new α-conotoxins with different pharmacological profiles can provide further insights into the physiological or pathological roles of the numerous nAChR isoforms found at the neuromuscular junction, the central and peripheral nervous systems, and other cells such as immune cells. This study focuses on the synthesis and characterization of two novel α-conotoxins obtained from two species endemic to the Marquesas Islands, namely *Conus gauguini* and *Conus adamsonii*. Both species prey on fish, and their venom is considered a rich source of bioactive peptides that can target a wide range of pharmacological receptors in vertebrates. Here, we demonstrate the versatile use of a one-pot disulfide bond synthesis to achieve the α-conotoxin fold [Cys 1-3; 2-4] for GaIA and AdIA, using the 2-nitrobenzyl (NBzl) protecting group of cysteines for effective regioselective oxidation. The potency and selectivity of GaIA and AdIA against rat nicotinic acetylcholine receptors were investigated electrophysiologically and revealed potent inhibitory activities. GaIA was most active at the muscle nAChR (IC_50_ = 38 nM), whereas AdIA was most potent at the neuronal α6/3 β2β3 subtype (IC_50_ = 177 nM). Overall, this study contributes to a better understanding of the structure–activity relationships of α-conotoxins, which may help in the design of more selective tools.

## 1. Introduction

Polynesia is an archipelago made up of over 100 islands spread over an area of 4 million km^2^. It is further divided into five archipelagos: the Society Islands, the Marquesas Islands, the Tuamotu Islands, the Gambier Islands, and the Austral Islands. Due to their geographical isolation, the islands of Polynesia have developed unique flora and fauna, with many endemic species found nowhere else in the world. Endemic cone snails of Polynesia are remarkable examples of this biodiversity, as these species are rare and unique to this region. These snails are characterized by their unique shells, which are often brightly colored and beautifully patterned. Among the most iconic are *Conus gauguini* (named after French painter Paul Gauguin) and *Conus adamsonii*, which are favorites among shell collectors. 

Nearly 1000 different species of these venomous marine mollusks have been described, and more than 20% are located in Polynesia [1]. They produce potent venom made up of hundreds of small peptides called conotoxins. These peptides can target a wide range of receptors, ion channels, and transporters. The cysteine framework, gene superfamily, and their pharmacological targets are used to classify the conopeptides [2]. Ziconotide, a synthetic version of the ω-conotoxin MVIIA, was isolated from *Conus magus* venom and approved by the FDA in 2004 for the treatment of chronic pain. It constitutes one of the very few drugs derived from marine natural compounds [3]. While ω-conotoxins block N-type calcium channels, the α-conotoxins target the nicotinic acetylcholine receptors (nAChRs), which are ion channels of primary physiological importance [4,5,6]. Indeed, nAChRs are widely distributed in the central nervous system (CNS) and the periphery, with several subtypes known to be involved in pain sensation, attention, learning, and memory. Thus, nAChRs are attractive pharmacological targets with potential applications in the treatment of pain, memory disorders, or Parkinson’s disease. To date, 16 homologous nAChR subunits have been identified in mammals (α1–7, α9, α10, β1–4, δ, and ε/γ), which can assemble in numerous combinations to form pentameric complexes of different nAChR subtypes [7,8,9]. The specific functions of each subtype are not fully understood, and α-conotoxins are among the most specific molecules that are capable of helping decipher their function [6,7,10,11,12].

In this work, we focused on the synthesis and functional characterization of two novel α-conotoxins, GaIA and AdIA, from the piscivorous cones *C. gauguini* and *C. adamsonii*, respectively. Both species are endemic to the Marquesas Islands. We applied a novel one-pot synthesis that leads directly to the native peptide. This was achieved by using the 2-nitrobenzyl (NBzl) protecting group in conjunction with the triphenylmethyl (Trt) group to ensure successful regioselective folding. This strategy was compared with the more commonly used oxidative folding strategies, which are based on thermodynamically driven folding of fully unprotected precursors or the use of the acetamidomethyl (Acm) and Trt protecting groups. Biological characterization using the two-electrode voltage clamp method revealed potent inhibitory activities at different nAChRs.

## 2. Results

### 2.1. α-Conotoxin Sequences

Two α-conotoxins, GaIA and AdIA, were retrieved from our in-house transcriptomic data on the piscivorous cones *C. gauguini* and *C. adamsonii* (to be published elsewhere), respectively. These conotoxins contain four cysteines and a predicted amidated C-terminus (Table 1). These conopeptides belong to different Cys loop spacing frameworks with varying m/n-loop sizes (m and n designate the number of non-cysteine residues between the conserved cysteine residues) identified in α-conotoxins, including α-3/5, α-3/7, α-4/5, α-4/7, and α-4/4. Interestingly, these frameworks appear to confer some subtype selectivity [11]. GaIA belongs to the α-3/5 sub-family, and AdIA belongs to the α-4/4 sub-family [13,14,15]. Since most characterized α-conotoxins belong to the 4/7 family and only comparably few 3/5 and 4/4 peptides have been synthesized and investigated before, we aimed to study their potency on specific nicotinic acetylcholine receptor subtypes (nAChRs). The 3/5 framework of GaIA generally affects muscle-type nAChR receptors [16,17,18], while the 4/4 framework of AdIA is of particular interest because it has been found to target both muscle-type and neuronal nAChR receptor subtypes [11]. AdIA appears very similar to BuIA, which targets several neuronal subtypes [19] and differs by only one residue from AdIA (BuIA has a tyrosine residue instead of a histidine residue in position 13). GaIA is most closely related to α-MI and α-GI, and all three have the following sequence in common: GXCCXPACGXXYXC (Table 1). A comparison of the potencies and sequences of these variants has the potential to reveal the molecular basis for the design of compounds with better selectivity for nAChR subtypes. From a chemical perspective, the presence of two neighboring proline residues in the sequence of AdIA represents a synthetic challenge due to potential folding difficulties. Furthermore, it might influence the overall structure and selectivity of the peptide, which makes the synthesis and study of these conopeptides a valuable contribution to the fields of both peptide synthesis and drug discovery.

### 2.2. Synthesis of Conotoxins

In order to compare the use of the Trt/Acm or Trt/NBzl pairs of cysteine protective groups for the preparation of α-conotoxins, we achieved the synthesis of the main precursors bearing a Trt at Cys II–IV and an Acm or NBzl at Cys I–III by Fmoc solid-phase peptide synthesis (SPPS) with an automated peptide synthesizer. The amino acid sequence of each precursor peptide is given in Appendix A. The initial linear peptides with a free thiol at positions II and IV were obtained after the cleavage of the peptide from the resin with similar yields and purity (Figure 1a and Figure 2a). Trt groups were removed during the cleavage step, while the Nbzl and Acm protective groups are resistant to the cleavage conditions described in experimental procedures. The first disulfide bridge was formed by treatment with disulfiram (DSF). While the high-performance liquid chromatography (HPLC) profile for the pair Trt/NBzl appears to be devoid of any impurity (Figure 1b and Figure 2b), the one obtained for GaIA with the Trt/Acm pair contains additional peaks (Appendix A). The second disulfide bonds were then formed by UV irradiation using NBzl as protecting groups or after successive treatment with PdCl_2_, DTC, and DSF for the Acm (Appendix A). All the protocols are detailed in the experimental procedures [20,21]. 

The GaIA sequence does not contain any amino acids that may constrain its three-dimensional structure. Therefore, both approaches led to the desired isomer with ease, as shown in Figure 1 and Appendix A. On the other hand, the AdIA sequence is unusual due to the presence of a Pro–Pro sequence, which allows for two different spatial configurations (Proline *cis* or *trans*). This particularity is interesting in terms of folding, and the *cis/trans* conformational dynamic equilibrium can be observed through the broadening of the peak during HPLC analysis at 24 °C, as depicted in Figure 2c (rt: 27.5 to 28.5 min). At lower temperatures (15.0 °C), we observed mostly a single conformer. The change in the *cis/trans* conformational equilibrium with temperature is shown in Appendix A. Similar HPLC peak broadening was also observed in the previously reported Pro–Pro containing the conotoxin BuIA [22]. 

Fully oxidized peptides were purified by RP-HPLC, and the molecular weights were confirmed using high-resolution mass spectrometry (HRMS) (Appendix A). Mass spectral analyses of fully oxidized GaIA showed a mass of 1561.640 Da after deconvolution, which is consistent with the expected theoretical mass (1561.6271 Da). Similarly, AdIA showed a mass of 1284.4890 Da after deconvolution, which matches the expected theoretical mass (1284.4871 Da). Overall, the final yields for the non-directed folding synthesis approach were approximately 7% for GaIA (for which only one isomer was obtained and hence a simpler purification step) and only 2–3% for the globular isomer of AdIA. In contrast, the “one-pot” method produced almost twice as high yields (13% for GaIA and 10% for AdIA via NBzl’s approaches). 

### 2.3. Functional Characterization

The inhibitory potency of AdIA and GaIA was investigated by two-electrode voltage clamp analysis on the following rat nAChR subtypes expressed in *Xenopus laevis* oocytes (Figure 3): the embryonic muscle-type nAChR (α1β1γδ), the homomeric neuronal α7 nAChR, the heteromeric α2β2, α3β2, α3β4, α4β2, α4β4 subtypes, and the α6/α3β4 and α6/α3β2β3 combinations, in which an α6/α3 chimera was used to reconstitute the respective α6 binding sites since α6 has been difficult to express [23] (Table 1). AdIA was most potent at the α6/α3β2β3 nAChR with a half maximal inhibitory concentration (IC_50_) of 177.3 nM but also inhibited α3β2 (IC_50_ = 1375 nM) with lower potency. No inhibition greater than 50% was observed for any of the other nAChRs at concentrations up to 10 μM. 

GaIA showed low nanomolar potency (IC_50_ = 38.37 nM) at the muscle-type nicotinic receptor α1β1γδ. However, it also inhibited the neuronal α3β2 nAChRs and the closely related α6/α3β2β3 nAChR with about 25-fold less potency (IC_50_ = 988.9 nM and IC_50_ = 1170 nM, respectively). IC_50_ values at the other neuronal nicotinic acetylcholine receptors were above 1 μM (α4β2: IC_50_ = 3931 nM, α7: IC_50_ = 5158 nM), α2β2: IC_50_ = 6474 nM, α3β4: IC_50_ = 7912 nM, α6/α3β4: IC_50_ = 18,160 nM, and α4β4: IC_50_ = 49,430 nM) (Table 2).

## 3. Discussion 

α-Conotoxins are venom components that are used by cone snails to aid in prey capture and defense against predators [24]. Due to their action on nAChRs, they also represent useful pharmacological probes to decipher their physiological roles and reveal the therapeutic potential of nAChR selective modulators. Indeed, nAChRs are ion channels involved in the modulation of neurotransmission in the central and peripheral nervous systems. The search for the most appropriate ligand for a given nAChR subtype has greatly benefited from the α-conotoxin chemiodiversity inherited from the evolution of *Conus* species in different environments [25,26]. Additionally, advancements in chemical synthesis techniques and pharmacological characterization [16,27] have further facilitated the exploration of these compounds for their pharmacological properties. In this project, we are taking advantage of two endemic species from French Polynesia, *Conus gauguini* and *Conus adamsonii*, which produce the α-conotoxin GaIA and AdIA, respectively. These two conotoxins have not yet been synthesized or isolated from crude venom, but they could be useful in furthering our knowledge of nAChRs. Similar to most α-conotoxins, GaIA and AdIA possess four cysteine residues (that are predicted to form two disulfide bridges), which gives rise to three possible conformers, but the Cys I–III and Cys II–IV connectivity allows for a globular fold, which is the one found in these naturally occurring peptides [28,29]. 

The synthesis of these peptides can be achieved using two concurrent approaches: one based on the oxidative folding of a linear precursor with all thiols free, and the other on the regioselective and stepwise formation of individual disulfide bridges. While the first approach is attractive because only two disulfide bridges need to be formed, the correct folding conditions need to be optimized to favor the globular isomer [28,29]. In addition, purification processes are time-consuming as the compound is usually diluted in large volumes to avoid the formation of interchain bridges. In this case, the random formation of disulfide bridges is thermodynamically driven but is also a sequence-dependent process in which the more stable isomer is generally formed. Therefore, work was initiated by Gyanda et al. [21] to address this issue, and 30 different oxidation conditions were tested on various types of α-conotoxins to determine the best conditions leading to the formation of the globular isomer. Considering this study, we applied a buffer system of 0.1 M NH_4_HCO_3_ pH 8 with reduced and oxidized glutathione (GSH 100 eq/GSSG 10 eq), suitable for the α3/5 GaIA linear precursor. Additionally, we obtained a folded peptide in 48–72 h (Appendix A). For α4/4 conopeptides such as AdIA, very few experimental conditions have led to the formation of the globular isomer [28]. In most cases, oxidative folding has led to the formation of the ribbon isomer (Cys I–IV, Cys II–III). Nevertheless, the use of a 50/50 mixture of 0.1 M tris(hydroxymethyl)aminomethane (Tris.HCl)/isopropanol (IPA) (50/50), pH 8, with GSH/GSSG (100/10) at room temperature led to the correctly folded isomer for AdIA. This correctly folded isomer accounted for 70% of the different isomers observed (as depicted in Appendix A). The use of an organic solvent, especially for peptides with hydrophobic side chains such as AdIA, enables the prevention of peptide self-aggregation, thereby promoting proper folding. To ascertain whether the Cys I–III and Cys II–IV linkages were formed using the previously described strategy based on random oxidative folding, we synthesized the ribbon isomer of AdIA regioselectively. Furthermore, we performed a comparison with the peptides obtained through a regioselective approach and oxidative folding (Appendix A).

In recent years, several regioselective strategies have been developed, whether in solution or on solid support [30,31,32]. The synthesis of disulfide bond peptides typically involves the use of more than two protective groups for cysteines and often requires several synthesis steps with purification stages that can negatively impact final yields. However, a new method has been recently introduced that offers a one-pot regioselective approach [21]. For both approaches, using either Nbzl or Acm as protective groups, the folded peptides are obtained in less than 30 min. It is worth noting that the use of DSF in this one-pot method provides an advantage, allowing for the rapid formation of disulfide bonds in less than ten seconds. Controlling the oxidation process requires maintaining the appropriate pH level. If the pH is below 5, the crude peptide will only partially form the S– bond. Conversely, if the pH exceeds 7.5, disulfide bridge opening and bond reshuffling to form regioisomers can be observed, as we can see with the presence of ribbon isomers in Appendix A.

Additionally, achieving proper folding for certain sequences, such as AdIA, can be challenging due to the presence of a Pro–Pro motif that creates a spatial conformation unfavorable for globular isomers. However, conducting the reaction at 37 °C can assist in such cases by causing further denaturation of the peptide chain, which ultimately facilitates the completion of the final bridge. In addition to the versatility of this method, there is also the fact that it improves the yield of synthesis compared to oxidative folding by nearly two folds.

In both cases, the yield of non-directed folding synthesis was approximately twice as low compared to the “one-pot” method via NBzl’s approaches (7% for GaIA and 2–3% for AdIA vs. 13% and 10%, respectively). Moreover, in challenging cases such as AdIA, the presence of the major isomer facilitates purification. Concerning the two “one-pot” strategies, the use of NBzl is more convenient as it requires less time and manipulation than the use of Acm, specifically for deprotection. In general, for most α- conotoxins, the synthetic strategy is based on the use of the Trt/Acm pairs. Often, this involves the use of I_2_ to deprotect the Acm group. It has been established that the employment of this particular reagent may give rise to back-alkylation, or excessive oxidation of sulfonic acid, which can cause further adverse effects [33]. 

To date, only a few α-4/4 subfamily α-conotoxins have been characterized, including BuIA and LvIC [13,14]. BuIA showed low to medium nanomolar potency at α6/α3β2β3 (IC_50_ = 0.26 nM) and α6/α3β4 (IC_50_ = 1.54 nM) as well as α3β2 (IC_50_ = 5.72 nM) and α3β4 (IC_50_ 27.7, Table 1). Selectivity for α6/α3β4 over α3β4 (IC_50_ = 1200 nM) and all other subtypes was obtained in the analog [T5A,P60]BuIA (IC_50_ = 58.1 nM) [34]. 

LvIC inhibited only α6/α3β4 with a micromolar potency (IC_50_ = 3.3 μM) but hardly inhibited other subtypes at concentrations up to 10 μM. Remarkably, both potency and specificity for the α6/α3β4 nAChR were significantly improved in [D1G,∆Q14]LvIC (IC_50_ = 19 nM) [14]. Interestingly, the closely related AdIA (=[Y13H] BuIA) shows selectivity for the α6β2 interface with an IC_50_ value of 177.3 nM at the α6/α3β2β3 combination (α3 is considered a structural subunit that does not form a binding site). In contrast, it shows potency that is at least 8-fold lower at α3β2 (IC_50_ = 1375 nM) and micromolar potency (>10 μM) at all other subtypes. This makes it the first 4/4 framework α-conotoxin selective for the α6β2 interface.

GaIA belongs to the α-3/5 subfamily and is closely related to α-MI and α-GI. These α-conotoxins have been widely studied for their high selectivity towards different interfaces of the muscle type and Torpedo nicotinic receptors. Likewise, the newly synthesized peptide GaIA shows nanomolar potency at the muscle-type α1β1γδ nAChR with a similar IC_50_ (38.37 nM) as α-MI (IC_50_ = 12 nM) and α-GI (IC_50_ = 20 nM). However, while α-MI and α-GI showed no effects on α2β2, α3β2, α3β4, α4β2, α4β4 and α7 receptors [32], GaIA remarkably exhibited also some potency at the α9α10 (IC_50_ = 777.2 nM), α3β2 (IC_50_ = 988.9 nM), the closely related α6/α3β2β3 (IC_50_ = 1170 nM, as well as the α4β2 (IC_50_ = 3931 nM), α7 (IC_50_ = 5158 nM), α2β2 (IC_50_ = 6474 nM), and α3β4 (IC_50_ = 7912 nM) neuronal nAChRs. 

Thus, AdIA and particularly GaIA display a wider pharmacological profile on nAChR subtypes compared to the other characterized members of this family and might provide a template for further structure–activity studies.

## 4. Materials and Methods

### 4.1. Abbreviations

Acm, acetamidomethyl; ACN, acetonitrile; DCM, dichloromethane; DMF, N,N-dimethylformamide; DIPC, disopropylcarbodiimide; DTT, dithiothreitol; DTC, diethyldithiocarbamate; DSF, disulfiram; eq, equivalent; ESI-MS, electrospray ionization-mass spectrometry; FA, formic acid; Fmoc, fluorenylmethoxycarbonyl; GSH, reduced glutathione; GSSH, oxidized glutathione; IPA, isopropanol; LC/MS, liquid chromatography/mass spectrometry; nAChRs, nicotinic acetylcholine receptors; NBzl, 2-nitrobenzyl; RP-HPLC, reversed-phase high-performance liquid chromatography; SPPS, solid-phase peptide synthesis; TFA, trifluoroacetic acid; TIS, triisopropylsilane; Tris, 2-amino-2-(hydroxymethyl)propane-1,3-diol; Trt, trityl; UV, ultraviolet. 

### 4.2. Chemical Synthesis

#### 4.2.1. General Procedure for the Synthesis of Linear Conotoxin Peptides

The linear precursors of conotoxins were synthesized on a Liberty Blue peptide synthesis instrument (CEM, Matthews, NC, USA) at a scale of 0.1 mmol. The synthesis was conducted on a Rink-amide resin with a loading capacity of 0.34 mmol/g. The standard amino acid coupling cycle is as follows: 2.5 mL of amino acid (5 eq) are added to the reactor, activated with 1 mL of disopropylcarbodiimide (DIPC) (5 equiv.) and 0.5 mL of Pure Oxyma (10 equiv.). The resulting mixture was heated at 90 °C in the microwave for 2.15 min. After each coupling reaction, the resin was subjected to a deprotection cycle of 2 min at 90 °C under microwave with a solution of DMF/Piperidine (80/20) followed by washing with DMF. Side-chain (except Acm and Nbzl) deprotection and cleavage from the resin were carried out using 10 mL of TFA/Tis/H_2_O/DTT (88/2/5/5) (*v*/*v*) under agitation for 2 to 3 h. The reaction mixture was filtered, and the filtrate was dropwise added to cold diethyl ether to give a crude peptide precipitate and centrifuged. This synthetic method was applied to all synthetic conotoxin precursors, which were obtained with correct yields.

#### 4.2.2. General Procedure for the Synthesis of Conotoxin via NBzl Protecting Group

The lyophilized linear conotoxin peptide was dissolved (0.5 mM) in 6 M Gn·HCl buffer, pH 7, and treated with 10 equiv. DSF for 10 s at 37 °C. Subsequently, the reaction mixture pH was adjusted to 6 using 0.1 M HCl and exposed to UV radiation at 350 nm (24 watts, ca. 3 × 10^16^ s cm^−3^ photons) in a photochemical 16 chamber reactor under ice-cooled conditions for 8–10 min. Purification was performed by using a semi-preparative C18 column.

#### 4.2.3. General Procedure for the Synthesis of Conotoxin via Acm Protecting Group

The lyophilized linear conotoxin peptide (AdIA or GaIA) was dissolved (0.5 mM) in 6 M Gn·HCl buffer, pH 7, and treated with 10 equiv. DSF for 10 s at 37 °C. Subsequently, the reaction mixture pH was adjusted to 1 using 0.1 M HCl, treated with 10 equiv. PdCl_2_, and incubated at 37.0 °C for 5 min. The reaction mixture was further treated with 30 equiv. of DTC (12 μL) followed by 10 equiv. of DSF, and the pH of the reaction was adjusted to 6.0 using 0.1 M HCl to complete the reaction.

#### 4.2.4. General Procedure for the Synthesis of Conotoxin via Oxidative Folding 

The lyophilized linear conotoxin peptide (0.25 mM) was dissolved in the selected buffer solution and subjected to agitation for 48 h. Upon completion of the reaction, the mixture was acidified to pH 3 using a 10% formic acid solution for subsequent analyses. Based on Gyanda et al. [28], the samples underwent UHPLC-HRMS analysis, and the peak areas were measured after ion extraction of the compound to determine the proportion of each isomer generated during the oxidative folding process. 

### 4.3. Mass Spectrometry

Analytical HPLC was performed on a Dionex Ultimate 3000 (Thermo Fisher Scientific, Waltham, MA, USA), using analytical XBridge Peptide BEH C_18_, 3.5 μm, Column (4.6 × 150 mm, 300 A) (Waters Corporation, Milford, MA, USA) at a flow rate of 1.2 mL/min, coupled with a diode array. All solvents used were HPLC-grade. 

The LC–MS system consists of a Thermo Fisher Scientific LC–MS device, an Accela HPLC coupled to a QFleet fitted with an electrospray ionization source, and an ion-trap analyzer. All analyses were performed using a Kinetex 2.6 μm C18 column (150 × 3.00 mm) from Phenomenex Inc. (Torrance, CA, USA) in linear gradient mode from 2% to 60% over 30 min with a flow rate of 0.5 mL/min (solvent A, water + 0.1% FA; solvent B, acetonitrile + 0.1% FA).

The UHPLC/HRMS system consists of a Vanquish UHPLC (Thermo Fisher Scientific, Waltham, MA, USA) coupled to a QTOF Maxis II mass spectrometer (Brucker Daltonics, Billerica, MA, USA), source electrospray ionization mode, ESI+. All analyses were performed using a bioZen™ 2.6 μm Peptide XB-C18 column (150 × 2.1 mm, 100 A) (Phenomenex, Torrance, CA, USA) in linear gradient mode from 2% to 50% over 50 min with a flow rate of 0.3 mL/min (solvent A, water + 0.1% FA; solvent B, acetonitrile + 0.1% FA) at 40 °C. Acetonitrile and formic acid RS for LC–MS (Carlo Erba, Val de Reuil, France); ultra-pure water from PURELAB Chorus 1 (ELGA Veolia, Lane End, UK).

### 4.4. Preparative RP-HPLC

Semi-preparative purification of cyclic peptides was performed using a Waters 1525 chromatography system fitted with a Waters 2487 tunable absorbance detector with detection at 214 nm and 254 nm. Purification was performed by eluting buffer A (H_2_O, 0.1% FA) into buffer B (acetonitrile + 0.1% FA). The column used was a GRACE Vydac C-18 column (250 × 10 mm, 5 μm) (Thermo Fisher Scientific, Waltham, MA, USA) at a flow rate of 3 mL/min.

### 4.5. Electrophysiology

Plasmids encoding the neuronal nAChR subunits (rat α2, α3, α4, α6, β2, β3, β4) were provided by Jim Patrick (Baylor College of Medicine, Houston, TX, USA) and subcloned in pNKS2. Rat muscle α1, β1, γ, and δ subunits in pSPOoD were provided by Veit Witzemann (MPI for Medical Research, Heidelberg, Germany). The α6/α3 chimera was generated in pNKS according to [35,36,37]. Rat α9 and α10 nAchR subunits in pBS KS(-) were provided by B. Elgoyhen and J. Boulter (The Salk Institut; San Diego, UCLA Department of Psychiatry and Biobehavioral Sciences, California) and were subcloned in pNKS2. The subunit ratio of α9/α10 was 3:1. Oocytes were obtained from EcoCyte Bioscience (Dortmund, Germany) or surgically extracted (Az. 2532.Vet_03-19-77) from female *Xenopus laevis* (Nasco, Fort Atkinson, WI, USA) and kept at the core facility animal models of the biomedical center of the LMU Munich (Az:4.3.2-5682/LMU/BMC/CAM) in accordance with the EU Animal Welfare Act. 

Functional experiments were performed as previously described by Giribaldi et al. [35]. Briefly, plasmids were linearized and cRNA synthesized using the mMessageMachine kit (Invitrogen, Thermo Fisher Scientific, Waltham, MA, USA). Fifty nanoliters of cRNA (0.1–0.5 μg/μL) were injected per oocyte with equal ratios of subunits. Oocytes were stored at 16 °C in sterile filtered ND96 (96 mM NaCl, 2 mM KCl, 1 mM CaCl_2_, 1 mM MgCl_2_, 5 mM HEPES, and pH 7.4) containing 5 μg/mL gentamicin. After 1–4 days, two-electrode voltage clamp recordings were performed at −70 mV with a Turbo Tec 05X Amplifier (npi electronic, Tamm, Germany) and CellWorks software (version 6.2.2). Electrode resistances were less than 1 MΩ, and currents were filtered at 200 Hz and digitized at 400 Hz. The recording solution ND96, with or without 100 μM ACh, was automatically applied via a custom-made magnetic valve system combined with a manifold mounted closely above the oocyte, thus allowing a fast (<300 ms) and reproducible solution exchange. Agonist pulses (2 s) were applied in 4-min intervals. Toxins were manually applied from a 10× stock solution in the 50-μL measuring chamber and preincubated for 3 min. Current responses were normalized to control responses before toxin application. GraphPad Prism (version 9.3.1) was used for data analysis, and a four-parameter logistic fit (Hill-fit) with plateaus constrained to 100% and 0% was used to generate dose–response curves. Oocytes from at least two frogs were used for each data point.

## Figures and Tables

**Figure 1 marinedrugs-21-00356-f001:**
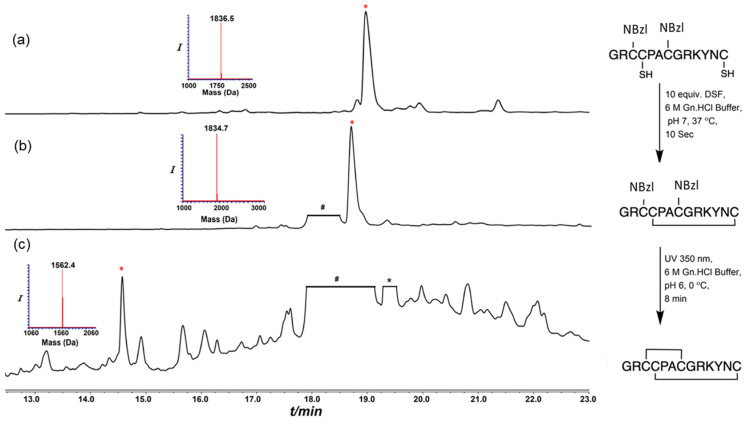
HPLC–MS analyses, GaIA synthesis via NBzl protecting groups; (**a**) the main peak observed corresponds to linear GaIA modified with NBzl at Cys I–III; (**b**) reaction after treating equiv. DSF, GaIA containing one disulfide bond; (**c**) reaction after UV radiation at 350 nm: conotoxin GaIA bearing two disulfide bonds. * non-peptide mass; * (red) desire product; and # non-peptide mass corresponding to DSF.

**Figure 2 marinedrugs-21-00356-f002:**
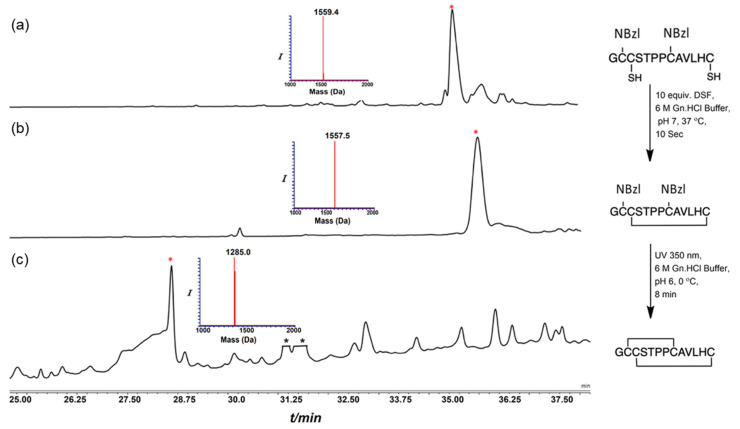
HPLC–MS analyses, AdIA synthesis via NBzl protecting groups; (**a**) the main peak observed corresponds to AdIA modified with NBzl at Cys I–III; (**b**) reaction after treating equiv. DSF, AdIA containing one disulfide bond; (**c**) reaction after UV radiation at 350 nm: conotoxin AdIA bearing two disulfide bonds. * non-peptide mass; * (red) desire product.

**Figure 3 marinedrugs-21-00356-f003:**
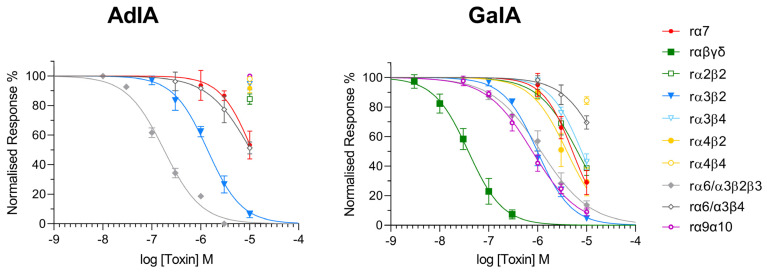
Dose–inhibition curves of AdIA (**left**) and GaIA (**right**) for the indicated rat nAChR subtypes expressed in *Xenopus laevis* oocytes. Two-electrode voltage clamp experiments were performed at −70 mV. Responses to 2-s pulses of 100 μM ACh were recorded after a 3-min preincubation with the indicated peptides. Each point represents the mean of at least three to four measurements with oocytes from at least two frogs.

**Table 1 marinedrugs-21-00356-t001:** Sequence alignment between α-conotoxin identified from Polynesian cone snails and previously reported α-conotoxins.

Framework Loop	Conotoxin	Title 3	Species
3/5	GaIA	GRCCHPACGRKYNC *	*Conus gauguini*
	MI	GRCCHPACGKNYSC *	*Conus magus*
	GI	ECCNPACGRHYSC *	*Conus geographus*
4/4	AdIA	GCCSTPPCAVLHC *	*Conus adamsonii*
	BuIA	GCCSTPPCAVLYC *	*Conus bullatus*
4/4	LvIC	DCCANPVCNGKHCQ	*Conus lividus*
	[∆Q14]LvIC	DCCANPVCNGKHC *	
	[D1G, ∆Q14]LvIC	GCCANPVCNGKHC *	

* C-terminally amidated.

**Table 2 marinedrugs-21-00356-t002:** IC_50_ values (in nM) of the conotoxins AdIA and GaIA on the indicated nAChR subtypes.

	AdIA		GaIA		BuIA	
nAChRs (Rat)	Toxin (IC_50_) nM	Hill Slope	Toxin (IC_50_) nM	Hill Slope	Toxin (IC_50_) nM	Hill Slope
α7	>10.000	N.D.	5158	−1.45	272	−1.21
			(4048 to 6680)	(−1.60 to −1.02)	(243 to 304)	(−1.10 to −1.32)
αβγδ	N.D.	N.D.	38.37	−1.23	N.D.	N.D.
			(32.44 to 45.40)	(−1.49 to −1.03)		
α2β2	>10.000	N.D.	6474	−1.09	800	−0.850
			(5296 to 8167)	(−1.39 to −0.83)	(567 to 1130)	(−0.591 to −1.11)
α3β2	1375	−1.24	988.9	−1.25	5.72	−1.48
	(1201 to 1574)	(−1.46 to −1.07)	(867.9 to 1128)	(−1.45 to −1.09)	(4.57 to 7.16)	(−1.04 to −1.92)
α3β4	>10.000	N.D.	7912	−1.37	27.7	−1.52
			(6638 to 9680)	(−1.76 to −1.06)	(22.3 to 34.5)	(−1.01 to −2.04)
α4β2	>10.000	N.D.	3931	−1.28	>10.000	N.D.
			(2578 to 6466)	(−1.86 to −0.71)		
α4β4	>10.000	N.D.	>10.000	N.D.	69.9	−1.15
					(47.9 to 102)	(−0.738 to −1.57)
α6/α3β2β3	177.3	−1.14	1170	−0.87	0.26	−0.963
	(153.3 to 205.4)	(−1.32 to −0.99)	(996.9 to 1376)	(−1.0 to −0.76)	(0.207 to −0.320)	(−0.815 to −1.11)
α6/α3β4	>10.000	N.D.	>10.000	N.D.	1.54	−1.40
					(1.32 to −1.78)	(−1.12 to −1.68)
α9α10	>10.000	N.D.	777.2	−0.93	N.D.	N.D.
			(679.8 to 889.3)	(−1.04 to −0.83)		
References	This work	This work	This work	This work	[13]	

95% confidence interval (CI) values are shown in parentheses. N.D. = not determined.

## Data Availability

Not applicable.

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
