# Peer review of "Synthesis and Biological Activity of Novel α-Conotoxins Derived from Endemic Polynesian Cone Snails"

_marinedrugs, 2023, doi:10.3390/md21060356_

Round 1

Reviewer 1 Report

The manuscript of Souf et al describes the synthesis of two novel α-conotoxins and their functional characterization towards different subtypes of nicotinic receptors. Both peptides were not detected in the gastropods’ venoms (the sequences were derived from transcriptomic data), the structure of both is close to the already described conotoxins and they did not find any superhigh potency to a distinct target and any unique selectivity profile to different nAChR subtypes. On the other hand, the studied compounds are derived from poorly studied representatives of the Conidae family and will replenish a large collection of already discovered α-conotoxins. And a preliminary characterization of their specificity in relation to different nAChR subtypes will allow them to be used for the designing new analogues. Taking into account these facts, this study can be published in the Journal after minor revisions according the comments made below.

Main points:

Synthetic part -

(lines 105-107) Table S1 presents precursors without Trt-groups, which formally contradicts the preceding phrase. For non-specialists in the field of SPPS, it is desirable to indicate in some place in the manuscript about the removal of these Trt-groups during cleavage of the resin.

It is advisable to indicate the obtained final yields of the target products (as one of the significant results) already in the Results section, especially taking into account the application of the new method with NBzl protections. The comparison of these yields with those from other methods is left for Discussion section.

It is also advisable to indicate the purity of the target products synthesized. It would be optimal to provide re-chromatographies of the target products in order to understand what purity conotoxins were used in functional tests. Judging by the figures shown (1c, 2c, S1, S2), the synthesized peptides do not seem to be sufficiently purified.

Functional part -

What explains the absence in a large series of tested nAChR subtypes the α9α10 one, which has a unique pharmacological profile?

The inhibition curve for AdIA on the muscle receptor in electrophysiological tests does not agree with the data presented in Table 2. The respective IC50 value, judging by the curve, is obviously less than 10000 nM and has a certain Hill coefficient. It would be nice to comment this “muscle-type” activity for AdIA in the Discussion.

In relation to the α7 receptor, it seems strange to accurately calculate the value of the Hill coefficient in the absence of one for IC50 in contrast to all other similar cases. In addition, observing a uniform style, it seems reasonable to show in Table 2 also a measurement error for the cited IC50 values of BuIA.

Minor corrections:

(line 73) Remove point after α-.

(lines 105, 232-236, 247) Decipher abbreviations - SPPS, GSH/GSSG, IPA, NBn and others - when they first appear in the text.

(line 150) Remove the slash between β2 and β3.

(line 166) Specify the number n by which the measurement error was calculated at each point.

(line 301) Add to the list of abbreviations – NBzl, GSH, GSSH, IPA…

Author Response

Point-by-point responses to the reviewer’s comments

We wish to thank both reviewers for their appreciation of our work and their suggested changes to improve our manuscript.

Reviewer 1

The manuscript of Souf et al describes the synthesis of two novel α-conotoxins and their functional characterization towards different subtypes of nicotinic receptors. Both peptides were not detected in the gastropods’ venoms (the sequences were derived from transcriptomic data), the structure of both is close to the already described conotoxins and they did not find any superhigh potency to a distinct target and any unique selectivity profile to different nAChR subtypes. On the other hand, the studied compounds are derived from poorly studied representatives of the Conidae family and will replenish a large collection of already discovered α-conotoxins. And a preliminary characterization of their specificity in relation to different nAChR subtypes will allow them to be used for the designing new analogues. Taking into account these facts, this study can be published in the Journal after minor revisions according the comments made below.

Main points:

Synthetic part -

(lines 105-107) Table S1 presents precursors without Trt-groups, which formally contradicts the preceding phrase. For non-specialists in the field of SPPS, it is desirable to indicate in some place in the manuscript about the removal of these Trt-groups during cleavage of the resin.

We agree with your comment. It's an oversight on our part. We have made changes to the text accordingly: lines 297-299 from "The cleavage of the resin was carried out using 10 ml of TFA/Tis/H2O/DTT (88/2/5/5) (v/v) under agitation for 2 to 3 h." to "Side-chain (except Acm and Nbzl) deprotection and cleavage from the resin was carried out using 10 ml of TFA/Tis/H2O/DTT (88/2/5/5) (v/v) under agitation for 2 to 3 h.”

It is advisable to indicate the obtained final yields of the target products (as one of the significant results) already in the Results section, especially taking into account the application of the new method with NBzl protections. The comparison of these yields with those from other methods is left for Discussion section.

We have now added this information on final yields at the end of the result section 2.2 synthesis of conotoxins, line 145-149: “Overall, the final yields for the non-directed folding synthesis approach was approximatively 7% for GaIA (for which only one isomer was obtained and hence a simpler purification step) and only 2-3% for the globular isomer of AdIA. In contrast, the "one-pot" method produced almost twice as high yields 13% for GaIA and 10% for AdIA via NBzl’s approaches.”

It is also advisable to indicate the purity of the target products synthesized. It would be optimal to provide re-chromatographies of the target products in order to understand what purity conotoxins were used in functional tests. Judging by the figures shown (1c, 2c, S1, S2), the synthesized peptides do not seem to be sufficiently purified.

We agree and purity of 95 to 97% (as determined using HPLC-HRMS) for the final products is now included in Figures S4 and S5.

Functional part -

What explains the absence in a large series of tested nAChR subtypes the α9α10 one, which has a unique pharmacological profile?

We have tested the maximum of nAChR subtypes in the limited time allocated to this project prior to submission. Then, during the evaluation of our manuscript, we could perform the additional α9α10 subtype, for which surprisingly GaIA showed significant affinity (submicroM), whereas AdIA was inactive as expected. These additional experiments and data on the α9α10 subtype have now been included in the updated Figure 3 and Table 2.

The inhibition curve for AdIA on the muscle receptor in electrophysiological tests does not agree with the data presented in Table 2. The respective IC50 value, judging by the curve, is obviously less than 10000 nM and has a certain Hill coefficient. It would be nice to comment this “muscle-type” activity for AdIA in the Discussion.

We apologize, there was a mix-up with the figures. We removed the muscle data in the final figure as these data were very preliminary and would have needed more measurements to confirm. A proper analysis was not possible in the given time.

In relation to the α7 receptor, it seems strange to accurately calculate the value of the Hill coefficient in the absence of one for IC50 in contrast to all other similar cases. In addition, observing a uniform style, it seems reasonable to show in Table 2 also a measurement error for the cited IC50 values of BuIA.

Again, apologies, wrong table.

Minor corrections:

(line 73) Remove point after α-. Thank you, removed.

(lines 105, 232-236, 247) Decipher abbreviations - SPPS, GSH/GSSG, IPA, NBn and others - when they first appear in the text. Thank you, done.

(line 150) Remove the slash between β2 and β3. Thank you, removed.

(line 166) Specify the number n by which the measurement error was calculated at each point

In the legend of Figure 3: “Each point represents the mean of at least three to four measurements with oocytes from at least two frogs.”

(line 301) Add to the list of abbreviations – NBzl, GSH, GSSH, IPA… Thank you, done.

Author Response

Point-by-point responses to the reviewer’s comments

We wish to thank both reviewers for their appreciation of our work and their suggested changes to improve our manuscript.

Reviewer 2

Alpha-Conotoxins are probes for the characterization of the various subtypes of nicotinic acetylcholine receptors (nAChRs). Identifying new alpha-conotoxins with different pharmacological profiles can provide further insights into the physiological or pathological roles of the nAChR isoforms. This study focuses on the synthesis and characterization of two novel alpha-conotoxins identified from two species Conus gauguini and Conus adamsonii. The authors demonstrate the versatile use of a one-pot disulfide bond synthesis to achieve the alpha-conotoxin fold [Cys 1-3; 2-4] for GaIA and AdIA, using the 2-nitrobenzyl (Nbzl) protecting group of cysteines for effective regioselective oxidation and compared the yield obtained via oxidative folding. The potency and selectivity of GaIA and AdIA against rat nicotinic acetylcholine receptors were investigated electrophysiologically. GaIA was most active at the muscle nAChR (IC50=38 nM), whereas AdIA was most potent at the neuronal α6/3 β2β3 subtype (IC50=180 nM). Overall, this study contributes to a better understanding of the structure-activity relationships of alpha-conotoxins that may help in the design of more selective tools. This study uses one-pot reaction for the synthesis of peptides with 2-disulfide bonds and showed that it has better yield than the thermodynamically driven undirected oxidative folding. While the former method saves time and effort (48-72 h vs 30 min and efforts of dealing with large volume), making it an attractive strategy. The study is well-written and organized and can be accepted for publishing after addressing the following concerns and adding some additional experimental details.

  1. How did the authors quantify the peptides GaIA and AdIA, given that GaIA has an aromatic residue and AdIA does not? Did they use the same quantification method for both peptides? Please provide the details of the quantification method used in the experimental section of the paper for reproducibility.

We apologize as indeed we did not mention the quantification of the two peptides. We used the same method and relied on analyses performed using HRMS by integrating the areas under the peaks to establish a quantitative report allowing us to determine the percentage of each peptide. We added lines 317-323 to describe the method used for that.

  1. How the G/R conformations were determined, was that from retention time of peptides from regioselective reaction?

To confirm the acquisition of the correct isomer, we have accurately synthesized the R and G isomers of each conotoxin (Globular: C I-III, C II-IV / Ribbon: C I-IV, C II-III). Subsequently, we have characterized them using LC-MS to determine the retention time of each isomer for comparison with oxidative folding. Taking your feedback into account, we feel that this information is important and have now also included Figure S6 (co-injection of R and G isomers of AdIA) to visually represent our findings.

  1. Did they test the R conformation of the peptides? R form also give rise to specificity as previously reported.

Unfortunately, no, we did not test the R conformation. In the limited time dedicated to this project, we decided to focus on the G conformations, as it is generally the native and most active fold. In the future, if we have the opportunity, we will investigate the activity of the R isomers.

  1. In the oxidative folding of AdIA, only the folding condition with IPA gives the correct folds (mostly G form), indicating the prevention of aggregation/precipitation can lead to correct folding. Please comment while discussing the comparative yield of both methods. Also, mention the ratio of GSH/GSSH.

Regarding the use of IPA as a co-solvent to promote proper isomer formation, this method had already been reported by Gyanda et al(1), explaining why we didn't initially include any comments about it. However, we agree with your suggestion to add a remark on this in "Line 214-216". The GSH/GSSH ratio was 100 eq/10eq, now mentioned lines 207-208.

  1. In fig. S6, why the retention time for the G form is different for oxidative folding vs regioselective folding? Also, why the oxidative folding condition which gave the desired G form (0.1 M Tris/IPA, GSH/GSSH, fig. S5 last panel) was not compared against regioselective approach?

The reviewer is correct. For this figure (S6>S7), the retention time of G was different because the analysis was not performed at the same column temperature. We have conducted the analysis at the same temperature and have now corrected this and changed the figure accordingly. We apologize for this confusion.

  1. The red asterisk denoting the desired product is off place (fig S1 and S2, presuming the major peak is the desired product)

Thank you, we have fixed it.

  1. Please mention the gradient used to obtain the chromatograms in Fig.1, as they are different from Fig S1 (if they have used two different systems, mention in section 4.3)

Yes, indeed the instruments used to obtain Figure 1 and Figure S1 are different, as well as the columns used. Figure 1 was obtained from analytical HPLC using the Thermo instrument (Dionex Ultimate 3000), while Figure S1 was obtained from the Thermo Fisher LC-Qfleet. We appreciate your comment and have taken it into consideration. We have also added this information lines 329-333 to address this difference in instrumentation. “LC-MS system consists of Thermo Fisher Scientific LC-MS device, Accela HPLC coupled to an QFleet fitted with an electrospray ionization source and a ion-trap analyzer. All analyzes were performed using a Kinetex 2.6 μm C18 column (150 x 3.00 mm) from Phenomenex Inc. (Torrance, CA, USA) in linear gradient mode from 2% to 60% over 30 min with a flow rate of 0.5 ml/ min (solvent A, water+0.1% FA; solvent B, acetonitrile+0.1% FA).”

  1. Fig.S2, it appears from the chromatograms that ACM protection had given a far superior product than 2-nitrobenzyl group, but why the authors chose 2-nitrobenzyl protection? Also, it looks like the desired peak has split at the top.

Regarding Figure S2, we chose NBzl because this technique requires less manipulation and is much easier to implement compared to the Acm protections. The Acm method initially involves adjusting the pH from 6 to pH 1 after the formation of the first disulfide bridge to allow for Acm deprotection without risking disulfide bridge reshuffling. Then, gradually raising the pH to 6 favored a certain predominant conformation of the Pro-Pro sequence (cis/trans proline) during the formation of the second bridge, resulting in a distinct peak of the desired product. On the other hand, with NBzl, we maintained the pH at 6.5 throughout the experiment, allowing us to observe different stages of the Pro-Pro motif conformation. This explains why the peak is broader with 2-nitrobenzyl protection.

  1. In fig. S1, on Acm protection (panel B), after Trt deprotection and oxidation of C I-III, crude has many minor peaks and unwanted products, but all of them disappear after the second disulfide bond formation. Why?

For this figure (S1, panel B), we worked on the crude peptide without purification after resin cleavage. There were still some impurities (peptides with amino acid deletions) that were not very significant compared to the desired linear peptide. After the addition of DSF, some impurities reacted with the DSF, but their masses were not at all close to our compound. Then, with the reaction with palladium, their intensity decreased proportionally, just like our desired conotoxin product.
